# Potential Nosocomial Infections by the Zika and Chikungunya Viruses in Public Health Facilities in the Metropolitan Area of Recife, Brazil

**DOI:** 10.3390/tropicalmed7110351

**Published:** 2022-11-04

**Authors:** Larissa Krokovsky, Duschinka Ribeiro Duarte Guedes, Fabiana Cristina Fulco Santos, Kamila Gaudêncio da Silva Sales, Daniela Anastácio Bandeira, Claudenice Ramos Pontes, Walter Soares Leal, Constância Flávia Junqueira Ayres, Marcelo Henrique Santos Paiva

**Affiliations:** 1Entomology Department, Aggeu Magalhães Institute, Oswaldo Cruz Foundation, Recife 50740-465, Brazil; 2Immunology Department, Aggeu Magalhães Institute, Oswaldo Cruz Foundation, Recife 50740-465, Brazil; 3Pernambuco’s Health Department, Recife 50751-530, Brazil; 4Department of Molecular and Cellular Biology, University of California, Davis, CA 95616, USA; 5Life Sciences Center, Agreste Academic Center, Federal University of Pernambuco, Caruaru 55002-970, Brazil

**Keywords:** nosocomial infection, arbovirus, vector surveillance, *Aedes aegypti*, *Culex quinquefasciatus*

## Abstract

Since 2015, the Dengue, Zika, and Chikungunya viruses gained notoriety for their impact in public health in many parts of the globe, including Brazil. In Recife, the capital of Pernambuco State, the introduction of ZIKV impacted human population tremendously, owing to the increase in the number of neurological cases, such as the Guillain–Barré and congenital Zika disorders. Later, Recife was considered to be the epicenter for ZIKV epidemics in Brazil. For arboviral diseases, there are some risk factors, such as climate changes, low socioeconomic conditions, and the high densities of vectors populations, that favor the broad and rapid dispersion of these three viruses in the city. Therefore, continuous arbovirus surveillance provides an important tool for detecting these arboviruses and predicting new outbreaks. The purpose of the present study was to evaluate the circulation of DENV, ZIKV, and CHIKV by RT-qPCR in mosquitoes collected in health care units from the metropolitan area of Recife (MAR), during 2018. A total of 2321 female mosquitoes (357 pools) belonging to two species, *Aedes aegypti* and *Culex quinquefasciatus*, were collected from 18 different healthcare units, distributed in five cities from the MAR. Twenty-three pools were positive for ZIKV, out of which, seventeen were of *C. quinquefasciatus* and six were of *A. aegypti*. Positive pools were collected in 11/18 health care units screened, with Cq values ranging from 30.0 to 37.4 and viral loads varying from 1.88 × 10^7^ to 2.14 × 10^9^ RNA copies/mL. Nosocomial *Aedes*- and *Culex*-borne transmission of arbovirus are widely ignored by surveillance and vector control programs, even though healthcare-associated infections (HAI) are considered a serious threat to patient safety worldwide. Although the results presented here concern only the epidemiological scenario from 2018 in MAR, the potential of hospital-acquired transmission through mosquito bites is being overlooked by public health authorities. It is, therefore, of the ultimate importance to establish specific control programs for these locations.

## 1. Introduction

Arthropod-borne viruses (arboviruses) are part of a large group of viruses that are transmitted to hosts (animals and humans) by hematophagous insects, such as mosquitoes and sand flies [1,2]. In the past years, various arboviruses gained notoriety for their impact on public health in different parts of the globe, such as the Yellow Fever (YFV), Dengue (DENV), Chikungunya (CHIKV), and Zika (ZIKV) viruses [3,4,5,6]. In 2015, Brazil faced a unique epidemiological scenario, with multiple reports of arbovirus-like symptoms and an enormous number of neurological cases (Guillain–Barré and congenital Zika syndromes). The Brazilian Northeast region was particularly affected and considered to be the epicenter of the disease, but Zika later spread to the rest of the country [7,8]. Within the Brazilian northeast region, Recife, the capital of Pernambuco state, was the most affected city in Brazil, with the highest number of congenital Zika syndrome cases than any other Brazilian capital [7,9]. In addition to ZIKV, DENV and CHIKV were simultaneously circulating in the city. Among several risk factors for arbovirus infections, the low socioeconomic conditions and the high densities of mosquito vectors found in Recife favored the broad and rapid dispersion of these three viruses in the area [9,10].

Dengue is the only one of these arboviruses possibly preventable with a vaccine. Still, the dengue vaccine is safe and prevents severe dengue in a subsequent infection, with long-term protection in seropositive individuals, while analysis revealed an excess risk of severe dengue in seronegative vaccinated individuals, compared to seronegative non-vaccinated [11,12,13]. In short, there are no effective and safe vaccines to control these arboviruses and no anti-viral treatments. Therefore, effective mosquito management programs are crucial for reducing mosquito populations, mosquito bites, and consequently, the transmission of arboviruses.

Although many characteristics are shared between both mosquito species found in the MAR, such as habitats, life-cycle patterns, and anthropophilic behavior, significant differences are found in the role as vectors of disease agents. *Aedes aegypti* has been identified as the main species responsible for the viral transmission dynamics of DENV, ZIKV, and CHIKV in different settings [14]. Although field-collected *Culex quinquefasciatus* have also been found positive for these arboviruses, vector competence studies do not corroborate some of these surveillance data [15,16].

From 2015–2017, extensive mosquito surveillance was conducted in the metropolitan area of Recife (MAR), during the triple epidemics of DENV, CHIKV, and ZIKV, to evaluate virus circulation in the urban mosquitoes *A. aegypti* and *C. quinquefasciatus* [17]. This surveillance, combined with vector competence studies, supported the role of *C. quinquefasciatus* in ZIKV transmission in Recife [18]. Later on, this data was corroborated by studies with virus isolation conducted in Florida (USA), Jalisco (Mexico), and Thailand [19,20,21]. These findings stress the importance of mosquito surveillance for monitoring arboviral transmission, particularly when (re)emergent virus activity in mosquitoes precedes human infections and can prevent possible outbreaks [22,23].

Epidemiological data collected from human cases in Pernambuco from late 2017 to early 2018 showed a significant reduction in the number of DENV, CHIKV, and ZIKV infections [24,25]. During that period, the arbovirus surveillance strategy shifted from randomly sampling mosquitoes to focusing on hotspots in the MAR. Here, we report our data suggesting that there is a high risk of nosocomial infections in hospitals in the MAR, Brazil. The results obtained in this study led to a One Health approach to control nosocomial infections in MAR, as it engaged researchers, public health professionals, government officials, and the general public.

## 2. Materials and Methods

### 2.1. Study Area

Throughout 2018, mosquitoes were captured monthly from January to December in major public hospitals in Recife, Brazil, and community health care clinics (UPAs) located in five different cities from the Metropolitan Area of Recife (MAR) (Figure 1, Table 1). Field collections were performed by the surveillance team from the Pernambuco’s Health Department (Secretaria Estadual de Saúde de Pernambuco/SES-PE).

### 2.2. Mosquito Sampling

Mosquito collections were performed during day hours (08:00 to 10:00 a.m. and 02:00 to 04:00 p.m.) using battery-operated aspirators (Horst Armadilhas Ltd., São Paulo, Brazil). Live mosquitoes were immediately transported in aspirator bags to the Entomology Department (FIOCRUZ-PE) in Recife, where they were cold anesthetized (−20 °C for 20 min), placed on a Petri dish on ice, and sorted by species, sex, location, date, and feeding status. The presence or absence of ingested blood in the abdomen was visualized using a stereomicroscope. After sorting, females were grouped into pools of up to ten individuals, according to the mentioned separation criteria, in DNAse/RNAse-free 1.5 mL microtubes and stored at −80 °C until further analysis. This study was approved by the Research Ethics Committees of the Aggeu Magalhães Institute (FIOCRUZ-PE) under the registration numbers CAAE 51012015.9.0000.5190 and PlatBr 1.547.598.

### 2.3. RNA Extraction and Standard Curve Synthesis

The following material was added to each mosquito pool: 300 µL of Leibovitz medium (L-15, Gibco, catalog #41300-039, Carlsbad, CA, USA) supplemented with 5% fetal bovine serum, 1% fungizon (Gibco, catalog #10270-106, Carlsbad, CA, USA), and antibiotics penicillin/streptomycin (Gibco, catalog #15140-122, Carlsbad, CA, USA). Mosquitoes were then homogenized with sterile micropestles. From this homogenate, 100 µL were aliquoted for RNA extraction using the TRIzol^®^ (Invitrogen, catalog #15596-026, Carlsbad, CA, USA) method with minor modifications [18], followed by Turbo DNAse (Ambion, catalog #AM2238, Foster City, CA, USA) treatment, according to manufacturer’s protocol. To detect and quantify DENV, CHIKV, and ZIKV in these mosquito pools, each sample was compared to a standard curve using an absolute quantification in reverse transcription quantitative real-time PCR (RT-qPCR) assays. The virus strains, maintained in Vero and C6/36 cells, ZIKV BRPE243/2015 (KX197192), CHIKV BRPE408, and DENV-2 3808/BR-PE (EU259569), were used as positive controls. Supernatant from each virus stock was submitted to RNA extraction, quantified in NanoDrop 2000 (Thermo Scientific, Waltham, MA, USA), and used as templates for an in vitro transcription using the MEGA script T7 kit (Ambion, Catalog #AM1333, Foster City, CA, USA), following the manufacturer’s protocol. After in vitro transcription, each sample was quantified in NanoDrop 2000, and RNA concentration was converted into RNA copy numbers, using the formula described by Kong et al. [26].

### 2.4. Optimization of the Multiplex RT-qPCR Assays

To set a RT-qPCR assay for a single multiplex reaction capable of simultaneously detecting three viruses, we used sets of primers and probes employed for DENV [27], ZIKV [28], and CHIKV [29] detection (Appendix A). Reactions were performed using the QuantiNova Probe RT-PCR kit (Qiagen, catalog #208354, Hilden, Germany). The mixture consisted of 0.08 µL of each primer (800 nM), 0.04 µL of each probe (100 nM), 5.0 µL of QuantiNova Probe RT-PCR Master Mix (5×), 0.1 µL of QuantiNova Probe RT Mix, 0.05 µL of ROX passive reference dye, and 3.5 µL of the transcripts dilutions, in a final volume of 10 μL. Cycling conditions were 15 min at 45 °C and 5 min at 95 °C, followed by 45 cycles of 5 s at 95 °C and 45 sec at 60 °C. Multiplex RT-qPCR assays were done on a QuantStudio 5 Real-Time PCR System (Applied BioSystems, Waltham, MA, USA), with automatic baseline and threshold. The singleplex reaction (separate for each virus) was carried out using the same PCR conditions and concentrations from the multiplex reaction, only adjusting the water volume.

### 2.5. Specificity and Sensitivity Analysis of the Multiplex Assays

First, each set of specific primers and probes for DENV, ZIKV, and CHIKV were tested to detect only the expected target. Standard curves were prepared using nine serial dilutions (10^12^ up to 10^4^ RNA copies/µL equivalent to 10^8^ fg up to 100 fg RNA concentration/µL), normalized in equal parts, mixed into a single microtube, and stored at −80 °C, and each reaction included a negative control. Analytical sensitivity was determined as the lowest amount of RNA detectable in a given reaction. Amplification efficiency (E) was calculated using the slope of the regression line in the standard curve, according to the equation: E = 10(−1/slope) − 1; a slope value close to −3.33 was considered satisfactory. The correlation coefficient (*R^2^*) value was automatically calculated using the measure of the strength of the relationship between the regression line and the individual Cq data points of the standard reactions. The y-intercept value was also automatically calculated and corresponded to the theoretical Cq value for a single copy of the target RNA.

Samples with cycle quantification (Cq) values of ≤38.5 were considered positive. The reactions were performed in technical triplicates and repeated three times for singleplex and multiplex reactions. In intra-assays, triplicates were performed on the same plate, whereas in inter-assays, triplicates were repeated in three independent assays.

### 2.6. Multiplex RT-qPCR Assay of Field-Collected Mosquitoes

After optimization, multiplex reactions were performed with mosquito samples: samples and controls were tested in duplicates. Controls comprised two different negative controls (derived from the RNA extraction and the RT-qPCR non-template control), as well as a positive control for each virus that was included in every 96-well RT-qPCR reaction plate. RT-qPCR assays were run on a QuantStudio 5 Real-Time PCR System (Applied BioSystems, Waltham, MA, USA), with automatic baseline and threshold. Samples that produced Cq values of ≤38.5 in both duplicates were considered positive. After that, positive samples were submitted to the second round of RNA extraction and RT-qPCR, including the standard curve to calculate the number of copies of viral RNA.

### 2.7. Data Analysis

Real-time RT-PCR results were analyzed using QuantStudio Design and Analysis Software 1.3.1 (Applied BioSystems, Waltham, MA, USA) and GraphPad Prism software v.8.02 (GraphPad, San Diego, CA, USA).

## 3. Results

A total of 2321 female mosquitoes (357 pools) belonging to two species, *A. aegypti* (712 specimens) and *C. quinquefasciatus* (1609 specimens), were collected from 18 different health care units, distributed in five cities in the MAR. Most of the *A. aegypti* pools (~90%) showed evidence of recent blood meal, whereas half of *C. quinquefasciatus* pools (~51%) showed abdominal distension produced by blood ingestion (Table 2).

Concerning the analytical sensitivity for both singleplex and multiplex RT-qPCR reactions for DENV, ZIKV, and CHIKV, detection limits were estimated at 2000 fg RNA/µL. The linear regression analysis of the standard curves confirmed the linearity of the singleplex reactions for DENV (*R*^2^ = 1.0, E = 89.40, slope = −3.61, y-intercept = 60.90), ZIKV (*R*^2^ = 1.0, E = 91.80, slope = −3.54, y-intercept = 62.75), and CHIKV (*R*^2^ = 1.0, E = 93.20, slope = −3.50, y-intercept = 61.50). Similarly, the linearity of the multiplex reaction for DENV (*R*^2^ = 1.0, E = 98.40, slope = −3.36, y-intercept = 59.909), ZIKV (*R*^2^ = 1.0, E = 96.61, slope = −3.41, y-intercept = 62.30), and CHIKV (*R*^2^ = 1.0, E = 95.12, slope = −3.45, y-intercept = 62.59) were confirmed (Appendix A and Appendix A).

The reproducibility of the multiplex RT-qPCR was assessed between and within runs, based on standard curves. The coefficients of variation of intra- and inter- assays were in the range of 0.20–2.59% and 0.05–1.93%, respectively. Overall, the difference between the Cq values of the intra- and inter-assay was ≤2, suggesting that the multiplex reaction is reliable.

For the arboviral detection of field-collected mosquitoes, from a total of 357 analyzed pools, ZIKV was detected in 23 pools collected in 11 out of 18 medical facilities. Cq values obtained by RT-qPCR reactions ranged from 30.0 to 37.4, and the number of virus RNA copies/mL ranged from 1.88 × 10^7^ to 2.14 × 10^9^ (Table 3). The majority of ZIKV-positive pools were from mosquitoes collected at Hospital das Clinicas, located at the Federal University of Pernambuco (western Recife), where a high number of mosquito specimens was collected throughout the year. In this location, six pools comprised engorged mosquitoes (three pools from each collected species) that were found to be positive for ZIKV. A single non-blood fed pool of *A. aegypti* was found to be positive for ZIKV in this particular hospital. In Hospital Getúlio Vargas, another major facility located in western Recife, two unengorged pools (one each species) and a single engorged *A. aegypti* pool were found to be positive for ZIKV (Table 3). All three pools consisted of single mosquitoes. ZIKV was also found circulating in mosquitoes from six community health care clinics (UPA), located in each of the five cities in the metropolitan area of Recife. Three pools of *C. quinquefasciatus* (one engorged and two unengorged) were found to be positive for ZIKV in UPA Olinda, whereas an engorged *C. quinquefasciatus* pool collected in UPA São Lourenço da Mata tested positive for CHIKV (Table 3).

## 4. Discussion

Natural transmission of arboviruses consists of a triad of the presence of virus-infected patients, competent vectors, and susceptible individuals. Hospital-like environments deal with two of these factors on a daily basis, and ideally, with the absence of mosquitoes. However, our study revealed high infestation levels of urban mosquitoes, *A. aegypti* and *C. quinquefasciatus*, in every public major hospital and community clinic that was screened in the MAR.

There was an overall high abundance of *C. quinquefasciatus* mosquitoes in every health care unit from the MAR. This fact alone is not surprising, since this species is nearly 20 times more abundant than *A. aegypti* in indoor areas from the MAR [30]. Likewise, collections performed during the day hours with aspirators favor the capture of *Culex* specimens, rather than *A. aegypti* [31,32]. A total of six *A. aegypti* pools were found to be positive for ZIKV, of which four were composed of engorged females. Since no other mosquito collection methodology was used in the health care units, and all collections were performed during daytime visits, the number of mosquitoes and arbovirus circulation in this species may be underestimated.

Studies conducted with mosquitoes collected in Recife provided strong evidence that *C. quinquefasciatus* could transmit ZIKV under laboratory and field conditions [18,33]. From over 100 screened pools of field-collected *C. quinquefasciatus*, Guedes et al. (2017) [18] found at least two ZIKV-positive samples that were not derived from a recent meal. Data obtained here showed that 17 pools of *C. quinquefasciatus* were positive for ZIKV, and six of these exhibited no evidence of blood meal, indicating that the virus was replicating in the mosquito, rather than being recently acquired by hematophagy. Similar data was reported by Krokovsky et al. (2022) [17], who screened 549 pools (~2500 *C. quinquefasciatus*) collected in Recife and its Metropolitan Area, and found ZIKV in 49 polls of non-engorged females. A vector surveillance conducted in Vitória, Espirito Santo State (Southeastern Brazil), also showed the presence of ZIKV in field-caught *C. quinquefasciatus* mosquitoes. All the ZIKV-positive pools comprised non-blood-fed females [30]. In a recent survey conducted in six regions in Thailand, Phumee et al. managed to detect ZIKV RNA in *C. quinquefasciatus* samples. Although epidemiologically significant, these results may be overestimated, as some of these samples were blood-fed [20].

Despite the fact that one CHIKV-infected *C. quinquefasciatus* sample was found in a health clinic in São Lourenço da Mata, this pool was composed of engorged mosquitoes, suggesting that these individuals had recently fed on viremic humans. Since evidence shows that this species is not able to transmit CHIKV [20], this result can only highlight the hyperendemicity found in that area. Similar findings were reported by Cruz et al. (2020) [6] in Brazil and Lutomiah et al. (2021) [34] in Kenya, who reported CHIKV-positive when analyzing whole-body *C. quinquefasciatus* derived from field collections.

Nosocomial *Aedes* and *Culex*-borne arbovirus transmission is widely ignored by surveillance and vector control programs, even though healthcare-associated infections (HAI) are considered a serious threat to patient safety worldwide [35,36]. This is particularly troublesome, as pointed out by Garza-González et al. (2017) [37], who reported a rapid ZIKV infection in pregnant women in a teaching hospital in Mexico. According to the WHO [38], the precise burden of HAI in low and middle-income countries remains undetermined. Although the results reported here concern only the epidemiological scenario in 2018 in MAR, the potential of nosocomial transmission through mosquito bites is being overlooked by public health authorities. It is, therefore, of the utmost importance to eliminate mosquito breeding sites and establish a specific program for these areas.

During the present study, bi-monthly meetings were conducted with Pernambuco’s State Health authorities. These meetings aimed to provide “real-time” data and enable the design of a One Health approach-based program, with the aim of implementing control measures exclusively for health care facilities, such as The Hospital das Clínicas, a major health care and teaching facility located on the Federal University of Pernambuco campus. During the first semester of 2018, after presenting ongoing results to hospital managers, a conjoint inspection, conducted on the premises, found mosquitoes circulating from the ground to upper floors, as well as a variety of breeding sites in the basement and around the building. This One Health approach encouraged hospital managers, local public health authorities, and researchers to design adequate control measures and routinely conduct surveillance in the hospital. In a similar strategy, Almeida-Nunes et al. (2016) [39] described four *Aedes*-transmitted dengue cases, nosocomially acquired in a major public hospital in São Paulo. Although these authors did not explore the infection status of the mosquitoes, the observation of these vectors and breeding sites in different areas led to specific control measures for the hospital environment [36]. Cotteaux-Lautard et al. (2013) [40] conducted a survey to monitor the local populations of *A. albopictus* in hospitals in two French cities and pointed out the necessity to consider solid protective measures in hospital-like environments, using the protection of detected human cases (repellent spray, bednets, and isolation) and setting up the physical control of mosquitoes in and around the grounds.

## 5. Conclusions

The prevention of the nosocomial transmission of arboviruses in the MAR is imperative, particularly considering the enormous number of asymptomatic ZIKV cases, the year-round activity *A. aegypti* and *C. quinquefasciatus* mosquitoes, and the hyperendemic characteristic of the MAR, where arbovirus cases account for almost 30% of the demand in health care units. Although nosocomial arboviral infections have been broadly reported worldwide, this is the first study to show the circulation of arbovirus-infected mosquitoes in health care facilities, which contain the perfect combination of breeding sites and immunocompromised patients.

## Figures and Tables

**Figure 1 tropicalmed-07-00351-f001:**
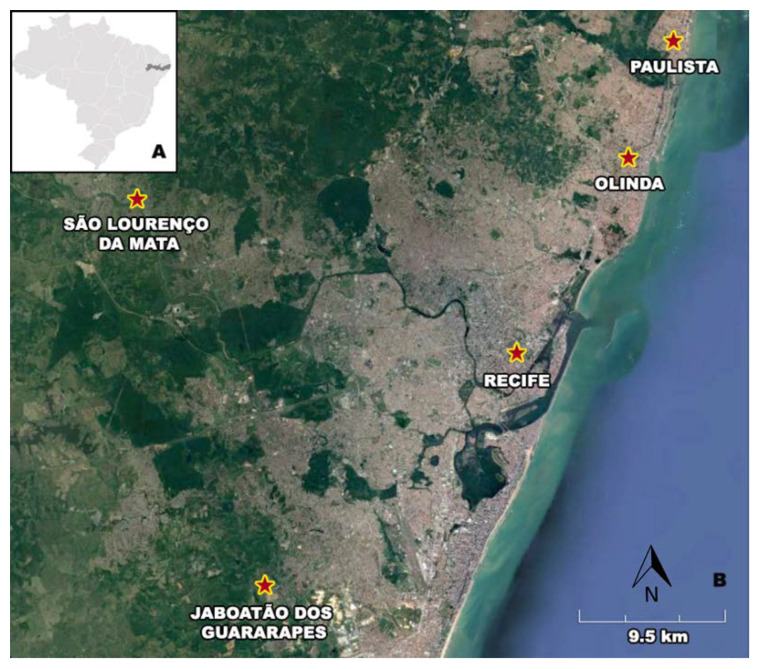
Geographical location and description of mosquito collection sites (red stars) in the metropolitan area of Recife. Legend: (**A**) Map of Brazil, highlighting Pernambuco state and (**B**) five cities from the MAR.

**Table 1 tropicalmed-07-00351-t001:** Location of health care facilities where mosquito collections were performed in Recife, Olinda São Lourenço da Mata, and Jaboatão dos Guararapes.

Health Care Facility	City	Coordinates
Hospital das Clínicas	Recife	−8.0476, −34.9461
Hospital da Restauração		−8.0538, −34.8978
Hospital Ulysses Pernambucano		−8.0332, −34.9022
Hospital Barão de Lucena		−8.0393, −34.9395
Hospital Agamenon Magalhães		−8.0304, −34.9075
Hospital Otávio de Freitas		−8.0871, −34.9615
Hospital Getúlio Vargas		−8.0512, −34.9217
Hospital Geral de Areias		−8.0100, −34.9265
UPA Curado		−8.0806, −34.9967
UPA Torrões		−8.0634, −34.9346
UPA Imbiribeira		−8.1207, −34.9137
UPA Caxangá		−8.0299, −34.9579
Secretaria de Saúde (FUSAM)		−8.0539, −34.8811
UPA Olinda	Olinda	−7.9710, −34.8661
UPA São Lourenço da Mata	São Lourenço da Mata	−7.9911, −35.0490
UPA Jaboatão dos Guararapes	Jaboatão dos Guararapes	−8.1109, −35.0067

**Table 2 tropicalmed-07-00351-t002:** Overview of specimens and blood-feeding status of mosquitoes sampled in health care units from the metropolitan area of Recife in 2018.

Period	Species	Number of Individuals	Number of Pools (Total)		Non-Blood Fed Pools	Blood Fed Pools
January	*A. aegypti*	16	(21)	3	1	2
	*C. quinquefasciatus*	78		18	9	9
February	*A. aegypti*	42	(23)	7	0	7
	*C. quinquefasciatus*	92		16	8	8
March	*A. aegypti*	42	(33)	8	0	8
	*C. quinquefasciatus*	141		25	11	14
April	*A. aegypti*	125	(46)	17	4	13
	*C. quinquefasciatus*	211		29	9	20
May	*A. aegypti*	33	(43)	9	2	7
	*C. quinquefasciatus*	325		34	28	6
June	*A. aegypti*	114	(32)	14	2	12
	*C. quinquefasciatus*	111		18	10	8
July	*A. aegypti*	43	(25)	6	0	6
	*C. quinquefasciatus*	84		19	8	11
August	*A. aegypti*	115	(23)	14	2	12
	*C. quinquefasciatus*	34		9	3	6
September	*A. aegypti*	21	(36)	5	1	4
	*C. quinquefasciatus*	163		31	14	17
October	*A. aegypti*	74	(29)	11	1	10
	*C. quinquefasciatus*	117		18	9	9
November	*A. aegypti*	41	(30)	6	0	6
	*C. quinquefasciatus*	183		24	9	15
December	*A. aegypti*	46	(16)	6	0	6
	*C. quinquefasciatus*	70		10	4	6
Total	*A. aegypti*	712	(357)	106	13	93
	*C. quinquefasciatus*	1609		251	122	129

**Table 3 tropicalmed-07-00351-t003:** Detailed characteristics from mosquito pools with detectable viral RNA loads from health care units from the metropolitan area of Recife (MAR).

Pool ID	Individuals Per Pool	Species	Collection Period	Healthcare Unit	Feeding Status	Viral Detection	Cq Mean Run 1	No of Copies Run 2
1640	10	*A. aegypti*	February	Hospital das Clínicas	BF	ZIKV	35.95	7.68 × 10^7^
1655	2	*C. quinquefasciatus*		UPA Curado	BF		35.75	1.88 × 10^7^
1666	10	*C. quinquefasciatus*		UPA Olinda	NBF		35.9	N.D.
1715	6	*C. quinquefasciatus*	April	Hospital Ulysses Pernambucano	NBF	ZIKV	36.15	7.82 × 10^8^
1803	10	*C. quinquefasciatus*	May	Hospital Barão de Lucena	NBF	ZIKV	33.85	3.38 × 10^7^
1813	1	*A. aegypti*		Hospital Getúlio Vargas	BF		34.75	N.D.
1814	1	*A. aegypti*			NBF		30.00	N.D.
1840	10	*C. quinquefasciatus*		UPA Olinda	BF		30.85	2.14 × 10^9^
1841	2	*C. quinquefasciatus*			BF		36.05	1.54 × 10^9^
1907	3	*C. quinquefasciatus*	July	Hospital Agamenon Magalhães	BF	ZIKV	36.10	6.21 × 10^7^
1939	2	*C. quinquefasciatus*		UPA São Lourenço da Mata	BF	CHIKV	31.90	2.19 × 10^7^
1948	10	*C. quinquefasciatus*	August	Hospital Otávio de Freitas	BF	ZIKV	30.2	N.D.
1966	1	*C. quinquefasciatus*		Hospital Getúlio Vargas	NBF		36.15	N.D.
2055	2	*C. quinquefasciatus*	October	Hospital das Clínicas	BF	ZIKV	34.60	N.D.
2064	3	*C. quinquefasciatus*		UPA Jaboatão dos Guararapes	NBF		35.95	N.D.
2071	9	*A. aegypti*		Hospital das Clínicas	BF		37.40	N.D.
2072	1	*C. quinquefasciatus*		UPA Torrões	NBF		35.15	N.D.
2111	10	*C. quinquefasciatus*	November	Hospital Ulysses Pernambucano	NBF	ZIKV	36.15	N.D.
2119	4	*C. quinquefasciatus*		UPA Paulista	NBF		36.30	N.D.
2156	10	*A. aegypti*	December	Hospital das Clínicas	NBF	ZIKV	37.00	1.95 × 10^7^
2160	10	*A. aegypti*			BF		35.10	N.D.
2164	10	*C. quinquefasciatus*			BF		36.75	N.D.
2165	10	*C. quinquefasciatus*			BF		35.80	N.D.

Note: BF, bloodfed; NBF, non-bloodfed.; N.D., non-detected.

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
