# Peer review of "Potential Nosocomial Infections by the Zika and Chikungunya Viruses in Public Health Facilities in the Metropolitan Area of Recife, Brazil"

_tropicalmed, 2022, doi:10.3390/tropicalmed7110351_

Round 1

Reviewer 1 Report

The authors have presented an interesting study of potential DENV, ZIKV, and CHIKV nosocomial infections in Recife, Brazil. The following comments should be addressed.

Abstract:

-          Line 33 Change Ae. aegypti and Cx. Quinquefasciatus to Aedes aegypti and Culex quinquefasciatus in first

Materials and Methods

-          The mosquito sampling part, the authors should explain briefly how to divide mosquito samples in a pool, such as 10 samples/pool, blood-fed, non-blood fed…...

-          Please include an animal ethics statement.

Results

-          Line 198, 200 italics of Ae. aegypti

-          Line 203, 205 italics of Cx. quinquefasciatus

-          Line 177 The authors mention singleplex and multiplex RT-qPCR methods. The method should be described in Materials and Methods.

Discussion

-          Line 236 Change Culex quinquefasciatus to Cx. quinquefasciatus

Conclusion

-          In general, the conclusion shouldn't include any citations. The conclusion needs to be revised.

References

-          Check all references, especially italicized species.

Author Response

REVIEWER 1 REPORT

Comments and Suggestions for Authors:
The authors have presented an interesting study of potential DENV, ZIKV, and CHIKV nosocomial infections in Recife, Brazil. The following comments should be addressed.

Abstract:
-          Line 33 Change Ae. aegypti and Cx. Quinquefasciatus to Aedes aegypti and Culex quinquefasciatus in first

Response: Thank you for pointing that out. We have italicized species names.

Materials and Methods:
-          The mosquito sampling part, the authors should explain briefly how to divide mosquito samples in a pool, such as 10 samples/pool, blood-fed, non-blood fed…... 

Response: Thank you. We have added an explanation about it in lines 104-107.

-          Please include an animal ethics statement. 

Response: Thank you. We have added a sentence about it (lines 108-110).

Results:
-          Line 198, 200 italics of Ae. aegypti

Response: Thank you. We have italicized Ae. aegypti in both lines.

-          Line 203, 205 italics of Cx. quinquefasciatus 

Response: We have italicized Cx. quinquefasciatus in both lines.

-          Line 177 The authors mention singleplex and multiplex RT-qPCR methods. The method should be described in Materials and Methods.

Response: We have added a description of the singleplex RT-qPCR in the Materials and Methods section (lines 142-144).

Discussion:
-          Line 236 Change Culex quinquefasciatus to Cx. quinquefasciatus 

Response: Thank you. We have italicized Cx. quinquefasciatus.

Conclusion:
-          In general, the conclusion shouldn't include any citations. The conclusion needs to be revised. 

Response: Thank you. We deleted such citations and reviewed the conclusion section.

References:
-          Check all references, especially italicized species. 

Response: Thank you. We checked all references, italicized species names and deleted a repeated citation.

Reviewer 2 Report

The mosquito-surveillance approach gives an indirect indication of the potential risk for human infection.  Data presented by the study showed a larger pool of Cx. quinquefasciatus infected mosquitoes compared to A.aegypt being the last moquitoe species a highly competent arbovirus vector.  It would be interesting to discuss potential differences between the two mosquitoes species and their competence to breed the three arbovirus and infect human hosts. 

The results point to the potential risk of nosocomial vector - dependent transmission but no data on the frequency of arbovirus infection related to nosocomial transmission was not shown.  Also, there was no citation of cases of hospital-acquired arboviruses in metropolitan area of Recife. Did the authors investigate the prealence of human infected population in the same geographical location of study area? How many cases were suspeted to be resulted from vector transmission inside the hospital areas?

Author Response

Thank you  very much for your instruction, the manuscript has been revised accordingly. The detail reply report is attached.

Reviewer 3 Report

Check over table 2. The October month data es not clear, apparently a Non-blood fed pool was not added

Author Response

Thank you very much for your instruction.

Please find the detail response in the attachment.

Reviewer 4 Report

Larissa Krokovsky et al evaluated the circulation of DENV, ZIKV and CHIKV by RT-qPCR in mosquitoes collected in health care units from the Metropolitan Area of Recife (MAR) 2018. Which showed that there was nosocomial Aedes and Culex-borne transmission of arbovirus in those hospitals.

1.      It is necessary to describe the time taken for each mosquito collection and collection time in Materials and Methods .

2.      The authors recorded that most of the Ae. aegypti pools (~90%) showed evidence of recent blood meal, whereas half of Cx. quinquefasciatus pools 173 (~51%) showed abdominal distension produced by blood ingestion in Results. So, it is necessary to describe how to find those evidences of recent blood meal in Materials and Methods.

3.      The authors should identify what species of animal blood were fed by engorged mosquitoes in the paper.

Author Response

(The authors gave the same response as above.)

Round 2

Reviewer 1 Report

NA

Reviewer 4 Report

The revised version have been sufficiently improved to publication.